# Modulation of Adverse Health Effects of Environmental Cadmium Exposure by Zinc and Its Transporters

**DOI:** 10.3390/biom14060650

**Published:** 2024-05-31

**Authors:** Ana Cirovic, Aleksandar Cirovic, Supabhorn Yimthiang, David A. Vesey, Soisungwan Satarug

**Affiliations:** 1Institute of Anatomy, Faculty of Medicine, University of Belgrade, 11000 Belgrade, Serbia; ana.zekavica@med.bg.ac.rs (A.C.); aleksandar.cirovic@med.bg.ac.rs (A.C.); 2Environmental Safety Technology and Health, School of Public Health, Walailak University, Nakhon Si Thammarat 80160, Thailand; ksupapor@mail.wu.ac.th; 3Centre for Kidney Disease Research, Translational Research Institute, Brisbane, QLD 4102, Australia; david.vesey@health.qld.gov.au; 4Department of Kidney and Transplant Services, Princess Alexandra Hospital, Brisbane, QLD 4102, Australia

**Keywords:** antagonism, bioavailability, cadmium, metal transporters, recommended dietary allowance (RDA), zinc

## Abstract

Zinc (Zn) is the second most abundant metal in the human body and is essential for the function of 10% of all proteins. As metals cannot be synthesized or degraded, they must be assimilated from the diet by specialized transport proteins, which unfortunately also provide an entry route for the toxic metal pollutant cadmium (Cd). The intestinal absorption of Zn depends on the composition of food that is consumed, firstly the amount of Zn itself and then the quantity of other food constituents such as phytate, protein, and calcium (Ca). In cells, Zn is involved in the regulation of intermediary metabolism, gene expression, cell growth, differentiation, apoptosis, and antioxidant defense mechanisms. The cellular influx, efflux, subcellular compartmentalization, and trafficking of Zn are coordinated by transporter proteins, solute-linked carriers 30A and 39A (SLC30A and SLC39A), known as the ZnT and Zrt/Irt-like protein (ZIP). Because of its chemical similarity with Zn and Ca, Cd disrupts the physiological functions of both. The concurrent induction of a Zn efflux transporter ZnT1 (SLC30A1) and metallothionein by Cd disrupts the homeostasis and reduces the bioavailability of Zn. The present review highlights the increased mortality and the severity of various diseases among Cd-exposed persons and the roles of Zn and other transport proteins in the manifestation of Cd cytotoxicity. Special emphasis is given to Zn intake levels that may lower the risk of vision loss and bone fracture associated with Cd exposure. The difficult challenge of determining a permissible intake level of Cd is discussed in relation to the recommended dietary Zn intake levels.

## 1. Introduction

Zinc (Zn) is a metal nutrient, which is required for the function of 10% of proteins in the human body and represents the second most abundant metal in the body, after iron (Fe) [1,2,3,4]. A total of 2–3 g of Zn is present in adults: 50–60% in muscles, 30–36.7% in bones, and 4.2%, 3.4%, and <1% in the skin, liver, and circulation, respectively [2,3,4]. The total plasma Zn concentration ranges between 9.9 and 24.5 µM (65–160 µg/dL), most of which is bound to albumin (80%) and the remainder to primarily α-2-macroglobulin, a protease inhibitor. The free plasma Zn concentrations are much lower and range between 1 and 3 nM [5,6,7].

Unlike Fe, the body does not have a storage mechanism for Zn, and thus, sufficient intake is required to maintain normal cell metabolism and function and prevent deficiency [4,5]. The recommended dietary allowance (RDA) for Zn is 8 and 11 mg/day for adult women and men, respectively [8]. However, it is noteworthy that the total plasma Zn concentrations do not reliably reflect the Zn body status, as it represents only a miniscule fraction of the total body Zn; there are significant diurnal variations, and there are large fluctuations in Zn levels with many disease states that include inflammation and infection [9,10,11]. Furthermore, Zn contributes to the specialized function of many tissues and organs, including bone turnover [12], hormones thyroxine and insulin [13,14], central nervous system development and function [15], immunity [16], and spermatogenesis [17].

Due to its wide range of biological roles, no specific symptom can be attributable to Zn deficiency, and it cannot be readily measured [2,3]. Thus, the health impact of marginal Zn intake and a subclinical Zn deficiency is not easily recognized. Inadequate dietary Zn intake is believed to be highly prevalent, especially in populations consuming diets low in Zn bioavailability, such as those that contain high amounts of phytate or are low in protein and calcium (Ca) [5,18,19,20].

Cadmium (Cd) is a metal contaminant with continuing public health concern worldwide because its presence disrupts the homeostasis and function of essential metal nutrients, including Zn, copper (Cu), Fe, Ca, and manganese (Mn) [21,22,23]. Dietary exposure to Cd is inevitable for most people as it is found in nearly all food types [24,25,26], polluted air, and tobacco smoke [27,28,29,30,31]. Children are particularly sensitive to the toxicity of airborne Cd, possibly due to the enhanced pulmonary uptake of Cd in this age group, especially in those with a small airway disease such as asthma [32,33,34]. Higher risks of having asthma were associated with lower plasma/serum Zn levels in a meta-analysis of 2205 children [35].

As there are no physiologic mechanisms for Cd elimination, continued exposure will lead to its accumulation in cells throughout the body. Primary public health measures should focus on minimizing exposure. A systematic review and meta-analysis showed that the body burden of Cd was inversely associated with body Fe stores and the nutritional status of Zn [36]. The Cd-Zn connection is also supported by the finding that elevated blood and urinary Cd levels are associated with particular genetic variants of the Zn transport proteins, ZIP8 and ZIP14 [37].

This review aims to provide insights into the roles of Zn and the impact of Cd exposure on health. Firstly, it summarizes the RDA for Zn together with current Cd exposure guidelines and its nephrotoxicity threshold level. It discusses the specific metal transporters which are responsible for Zn and Cd absorption and the cellular acquisition of these metals. Secondly, it discusses the correlation between Zn deficiency and Cd toxicity, and by interfering with Zn transporters, Cd perturbs Zn homeostasis. Thirdly, it summarizes the overall health threat linked with environmental non-workplace Cd exposure, evident from prospective cohort studies. The protective roles of Zn and its transport proteins against Cd-induced bone resorption are highlighted.

## 2. Zinc versus Cadmium

In this section, the RDA for Zn and the metal transporters responsible for its absorption are discussed together with a “tolerable” level of Cd. By definition, a tolerable intake level for a chemical with no known biological function is an estimate of the amount that can be ingested over a lifetime without an appreciable health risk [38].

The three main reasons for the persistent presence of cadmium in the environment are that over the years, it has been widely used in many industrial processes, it is a byproduct of many mining activities, and it is a frequent contaminant of cheap, low-grade phosphate fertilizers still used in developing countries [39,40,41,42].

The levels of Cd in foods are still likely to gradually increase over time because as a metal, it does not degrade, and it is readily taken up from the soil by food crops. Thus, whilst efforts should focus on reducing environmental contamination, ways to reduce the accumulation of Cd by plants and methods to enhance dietary Zn bioavailability are worth pursuing [43,44].

### 2.1. Recommended Dietary Allowance for Zinc versus Tolerable Level for Cadmium

The RDA values for Zn in various age groups and current dietary exposure guidelines for the tolerable intake levels of Cd are provided in Table 1.

#### 2.1.1. Zinc and Its RDA Values

According to the U.S. National Academy of Sciences Institute of Medicine (IOM), respective RDA values of Zn for an average adult 60 kg female and 70 kg male are 8 and 11 mg/day [8]. An additional 3 to 5 mg of Zn is required to meet increased physiological demands during pregnancy and lactating periods. The RDA for Zn may need to be increased further to help reduce the transfer of Cd to breast milk and its inhibitory effect on the secretion of Ca to breast milk, which has been observed in Cd-exposed Bangladeshi women [51,52].

#### 2.1.2. Tolerable Levels of Cadmium

According to the Food and Agriculture Organization and World Health Organization (FAO/WHO) Joint Expert Committee on Food Additives and Contaminants (JECFA), a tolerable intake level of Cd was set originally at 400–500 µg/person/week, which was later revised to 7 µg/kg body weight/week. Later, the weekly intake level was amended to a tolerable monthly intake of Cd at 25 μg per kg body weight per month, equivalent to 0.83 μg per kg body weight per day (58 µg/day for a 70 kg person), and the Cd excretion of 5.24 μg/g creatinine was used as a nephrotoxicity threshold level [38]. These figures were based solely on the excretion rate of β_2_M ≥ 300 μg/g creatinine as a toxic endpoint.

Like JECFA, the tolerable ingestion rates for Cd listed in Table 1 assumed the existence of a toxicity threshold level while using the urinary Cd excretion rate, normalized to creatinine excretion to reflect the long-term exposure or body burden of Cd [46,47,48]. As reviewed in Satarug, 2024, however, this method of the normalization of urinary Cd excretion created a high degree of statistical uncertainty, which obscured and nullified the quantification of Cd effects [53]. Consequently, none of the existing guidelines could reliably indicate a dietary intake level that carries a negligible health risk, inferred from the definition for a tolerable intake level of any contaminant.

The utility of urinary β_2_M excretion levels ≥ 300 µg/g creatinine, termed tubular proteinuria, as a toxic endpoint is another serious conceptual flaw. This endpoint is associated with severe kidney pathologies, resulting from tubular cell injury and death and malfunction and nephron destruction by Cd accumulation (Figure 1).

In phase 1, a submicroscopic tubular cell perturbation induced by Cd is evident from the appearance of KIM1 in the urine [22]. In Taiwanese patients with CKD, urinary Cd concentrations correlated with KIM1 but not other conventional renal pathologic biomarkers [54]. Thus, KIM1 excretion could serve as an early warning sign of Cd toxicity.

In phase 2, the GFR begins to decline as tubular cell damage and death are intensified with continuing Cd influx [55]. In phase 3, albuminuria and tubular proteinuria ensue because of defective reabsorption and a substantial loss of functioning nephrons.

Phase 1 appears to be reversible through cellular repair mechanisms if there is no further influx of Cd. The cessation of exposure may delay progression to phase 3, which is irreversible and can advance to end-stage kidney disease, when dialysis is necessary for survival.

#### 2.1.3. Loss of GFR as a Sensitive Toxic Endpoint

In theory, an exposure level that is more likely to produce discernable health risk needs to be based on the most sensitive endpoint [56]. Kidney tubular epithelial cell damage and death, depicted in Figure 1, are most frequently reported as signs of the nephrotoxicity of Cd in non-occupationally exposed conditions. However, a declining eGFR is a common sequela of ischemic acute tubular necrosis and acute and chronic tubulointerstitial fibrosis, all of which create impediments to filtration such as the destruction of post-glomerular peritubular capillaries, amputation of glomeruli from tubules, and obstruction of nephrons with cellular debris [57,58].

When GFR loss was used as a toxic endpoint, the NOAEL equivalent value or a toxicity threshold level of urinary Cd excretion ranged between 0.01 and 0.02 µg/g creatinine [45,59], which is at least 250 times less than 5.24 µg/g creatinine used by JECFA to derive a tolerable intake level for Cd [38]. This NOAEL figure implies that there is no safe level of Cd exposure.

In the China Health and Nutrition Survey (n = 8429), in which 641 participants (7.6%) had chronic kidney disease (CKD), the likelihood of having CKD increased 1.73-, 2.93-, and 4.05-fold when dietary Cd exposure rose from 16.7 to 23.2, 29.6, and 36.9 μg/day, respectively [60]. A dietary Cd exposure level of 23.2 µg/day is 40% of the JECFA tolerable exposure guideline, and yet it was associated with a 1.73-fold increase in the risk of the eGFR to fall below 60 mL/min/1.73 m^2^. This is not a negligible health risk.

#### 2.1.4. Summary of “Tolerable” Level of Cadmium

Current dietary Cd exposure guidelines range between 0.21 and 0.83 µg/kg body weight per day (Table 1). These guidelines assume Cd excretion rates of 0.5–5.24 µg/g creatinine as the threshold level for toxicity to kidneys or bones. However, the NOAEL equivalent value of urinary Cd excretion of 0.01 and 0.02 µg/g creatinine was obtained when a declining eGFR was employed as a toxic endpoint. It is now apparent that all existing tolerable dietary Cd intake levels are not low enough to protect human health due to exposure to Cd. Also, it should be noted that most or all excreted Cd originates from injured or dying kidney tubular epithelial cells. Thus, the excretion of Cd itself quantifies the severity of the kidney injury due to Cd accumulation at the present time, not the risk of injury in the future. We therefore argue whether a “tolerable” intake of a cumulative toxicant like Cd is a scientifically valid concept. At the very least, exposure to environmental Cd has been found to enhance the mechanisms underlying cellular senescence, involving surtuin-1 (SIRT1) [61,62,63].

### 2.2. Absorption of Metal Nutrients versus Contaminant Cadmium: Overview

Living organisms cannot synthesize or destroy any metals, and transport proteins have consequently been evolved to obtain all their required metals from an external environment. Generally, for humans, the intestinal absorption of metals from the diet involves two main processes: firstly, the transport of metals into enterocytes, mediated by influx transporters, localized to the brush border membrane, and secondly, the transport of metals to the basolateral membrane, where metals are extruded from enterocytes to portal blood, mediated by efflux transporters.

Table 2 lists the influx and efflux transporters that the body uses to acquire metal nutrients, Zn, Fe, Mn, Cu, and Ca.

#### 2.2.1. Dietary Zinc Absorption

Through highly specific transporters ZIP4, ZnT5, and ZnT1 [64,65,66,67,68,69,86], approximately 2.5–3.5 mg of Zn is absorbed to meet daily physiological requirements in adults, [5,9,10,18]. Phytate (myo-inositol hexabisphosphate), Ca, and proteins are three main food components that affect dietary Zn bioavailability [5]. Phytate chelates Zn, and the Ca fortification of phytate-rich soy milk has been shown to overcome the low Zn availability. By preventing Zn-Ca phosphate coprecipitation, caseinophosphopeptides increase Zn bioavailability in phytate-rich diets [87]. Protein intake may increase Zn absorption through Zn–amino acid co-transport mechanisms [5]. Intestinal absorption may also be enhanced by Zn ionophores [44,88].

Zn supplementation has many challenges, given the co-existence of Fe and Zn deficiencies and their interactions if supplemented together [4,5,20,77,89]. A high-dose Zn supplement necessitates an additional 2 mg of Cu to prevent anemia due to Cu deficiency induced by a high dose of Zn (≥80 mg/day) [5,90]. Decreased ZnT5 and ZIP4 protein levels were observed in ileal biopsies collected from volunteers, who consumed 25 mg of Zn as zinc sulfate with food daily for two weeks [67]. Increased systemic blood pressure and reduced kidney function, measured by inulin clearance, were observed in rats fed with a diet containing 40 times higher Zn than in a normal diet for 4 weeks [91].

#### 2.2.2. High Absorption Rate of Cadmium: Role of Multiple Transporters

As listed in Table 2, dietary Cd gains access to the systemic circulation and reaches cells throughout the body through several metal transporters, including those for Fe, Zn, Cu, and Ca. Furthermore, Cd complexed with MT and phytochelatin can be absorbed through transcytosis and receptor-mediated endocytosis [92,93,94]. Studies from Japan reported the absorption rates of Cd among women to be as high as 24–45% [95,96].

It is noteworthy that Cd absorption rates of 3–7% were used in the estimation of a tolerable level of Cd by JECFA (Table 1). Such an assumption of low intestinal Cd absorption rates led to the miscalculation of a tolerable exposure level of dietary Cd.

Cd can reach most cells in the body in the same way as metal nutrients, Zn, Fe, Ca, and Cu. Indeed, the cellular acquisition of Cd appears to be through many more transporters than those that mediate Cd entry into the enterocytes [97,98]. Of note, however, no excretory mechanism has been found for Cd. ZnT1 mediates the efflux of Zn but not Cd [68,69]. Similarly, FPN1 mediates the efflux of Fe, Zn, and Co but not of Cu, Cd, or Mn [77,78]. Consequently, most Cd acquired is retained within the cell after its entrance. In the absence of any efflux transporters for Cd removal exploitation, the avoidance/minimization of exposure is an essential measure to prevent many health threats, elaborated in Section 4.

## 3. Zinc and the Cytotoxicity of Cadmium

Zn is a type 2 nutrient that plays a role in fundamental biological mechanisms, namely protein and DNA syntheses, gene transcription and cell proliferation, division, and differentiation [3,77,99]. The maintenance of normal intermediary metabolism and cell function by Zn is accomplished through its role in cellular redox signaling and antioxidant defense mechanisms [99,100,101,102].

### 3.1. Zinc Homeostasis and Its Antioxidative Function

In cells, half of Zn is present in the cytoplasm, where it is bound to many proteins, and sequestered into subcellular organelles such as mitochondria, the endoplasmic reticulum (ER), vesicles, and the Golgi [103,104,105]. The cytosolic labile Zn concentration is in a picomolar-to-low nanomolar range, which is very low compared to the total intracellular Zn concentrations of 10–100 μM [103].

The cellular acquisition of Zn, its levels in cells, and trafficking between subcellular organelles are regulated tightly by specialized transport proteins, SLC30A (ZnT) and SLC39A (ZIP) [103,104,105]. These transport proteins are fundamental to Zn homeostasis. Zn itself also contributes to its homeostasis through the induction of ZnT1 [106] and MT to which 5–15% of cytosolic Zn is bound [107]. ZnT1 is a unique efflux transporter that functions as a Zn/Ca exchanger, which protects against a rise in cellular Zn [108,109].

In normal intermediary metabolism, including mitochondrial ATP synthesis, reactive oxygen species (ROS) are produced, and antioxidative mechanisms have been evolved to protect against oxidative damage, due to excessive ROS [100,101,102,110].

An overview of the molecular entities responsible for Zn homeostasis and its antioxidative function is presented in Figure 2.

The role of Zn as the modulator of the activity of superoxide dismutase (SOD) and NADPH oxidase (Nox) is well known [111,112]. The antioxidative function of Zn through heme biosynthesis and enhancing heme degradation and bilirubin synthesis has recently been demonstrated, detailed below.

The activity of δ-aminolevulinic acid dehydratase (ALAD), an enzyme in heme biosynthesis, is Zn-dependent. Zn deficiency anemia is caused by an insufficient amount of heme for hemoglobin synthesis [113]. Similarly, anemia due to lead (Pb) poisoning is attributable to decreased heme biosynthesis because of the Pb displacement of Zn in ALAD [114]. The activity of ALAD could also be decreased by Cd through its induction of MT, which eventually affects the formation of both heme and bilirubin.

A methodological breakthrough in measuring bilirubin in cells was made, following the discovery of a protein capable of binding unconjugated bilirubin [115]. Subsequently, it was demonstrated that heme was synthesized de novo in most cells for the continuous production of bilirubin, a cytoprotective biomolecule [116]. Bilirubin is a potent antioxidant and a lipid peroxidation chain breaker [117]. The discovery of Zn as an inducer of HO-1 accentuates further the role of Zn in bilirubin synthesis [118]. A reporter gene assay showed that Zn activated HO-1 gene expression via *antioxidant response element* (*ARE*) and the nuclear factor (erythroid-derived 2)-like 2 (Nrf2) signaling pathway [118].

### 3.2. Cadmium-Induced Disruption of Cellular Zinc Homeostasis and Redox State

More than a decade ago, Moulis (2010) published the hypothesis that the disruption of the homeostasis of physiologically required metals is a central toxic mechanism of Cd [119]. Because the ionic radius and electronegativity of Cd are close to Zn and Ca, the homeostasis of Zn and Ca is the primary Cd toxicity target [120,121].

An Australian autopsy study has provided evidence for the impacts of Cd exposure on Zn and Cu levels in livers and kidneys [23]. Similar effects of Cd on hepatic and renal Zn and Cu levels have been observed in rats 6 months after exposure cessation [122].

By the same way as Zn, Cd induces the expression of MT and ZnT1 simultaneously through its interaction with the metal response element-binding transcription factor-1 (MTF-1) [107]. Satarug et al. (2021) used UROtsa cells, a cell culture model of human urothelium [123], to explore the effects of acute exposure to Cd on the ZnT and ZIP expression profiles (Table 3).

**Table 3 biomolecules-14-00650-t003:** Changes in ZnT and ZIP transcript levels in UROtsa cells after exposure to cadmium.

Transporters	Transcripts Per 1000 β-Actin				
	Batch I, 0 µM Cd^2+^	Batch II, 0 µM Cd^2+^	1 µM Cd^2+^	2 µM Cd^2+^	4 µM Cd^2+^
**SLC30A**					
**ZnT1**	181 ± 23	365 ± 38	3007 ± 465	1434 ± 146	1216 ± 153 ***
**ZnT2**	0.01 ± 0.001	0.06 ± 0.01	73 ± 15	16 ± 1.9	11 ± 1.5 ***
**ZnT3**	0.03 ± 0.007	0.15 ± 0.01	0.24 ± 0.05	0.10 ± 0.02	0.10 ± 0.02 *
**ZnT4**	1.6 ± 0.26	11.4 ± 0.8	10 ± 1	8.7 ± 1	6.2 ± 0.5 **
**ZnT5**	150 ± 19	510 ± 30	1038 ± 132	495 ± 54	568 ± 91 **
**ZnT6**	4.5 ± 0.15	65 ± 8	77 ± 6	63 ± 13	57 ± 12
**ZnT7**	734 ± 28	758 ± 76	1007 ± 136	706 ± 44	488 ± 63 *
**ZnT10**	0.04 ± 0.005	1.1 ± 0.2	2.4 ± 0.2	1.7 ± 0.2	1.1 ± 0.1 ***
**SLC39A**					
**ZIP1**	19.5 ± 2.0	82 ± 9	99 ± 15	55 ± 10	59 ± 12 *
**ZIP2**	0.02 ± 0.004	1.2 ± 0.1	0.8 ± 0.2	0.4 ± 0.1	0.2 ± 0.03 ***
**ZIP3A**	9.1 ± 0.4	19 ± 1	23 ± 2.3	17 ± 1.7	14 ± 1.3 *
**ZIP3B**	0.48 ± 0.05	4.1 ± 0.2	6.2 ± 0.7	4.4 ± 0.2	4.2 ± 0.4 *
**ZIP4**	0.18 ± 0.04	0.06 ± 0.01	0.06 ± 0.01	0.04 ± 0.01	0.06 ± 0.01
**ZIP5**	0.01 ± 0.003	0.01 ± 0.001	0.01 ± 0.003	0.01 ± 0.002	0.01 ± 0.002
**ZIP6**	18.3 ± 2.6	92 ± 8	133 ± 12	80 ± 9	75 ± 10 **
**ZIP7**	121 ± 9.5	204 ± 25	342 ± 69	149 ± 32	94 ± 21 ***
**ZIP8**	0.09 ± 0.01	2.1 ± 0.2	2.6 ± 0.3	2.0 ± 0.2	2.7 ± 0.4
**ZIP10**	5 ± 0.3	54 ± 4	30 ± 8	14 ± 3	14 ± 3 ***
**ZIP14**	83.4 ± 10.5	146 ± 19	218 ± 24	158 ± 26	128 ± 19 *

The number of transcripts for each transporter was expressed per 1000 transcripts of the β-actin gene [123]. * *p* = 0.01–0.05; ** *p* = 0.001; *** *p* ≤ 0.001.

At the basal state, ZnT7 was the most abundantly expressed among the ZnT family members, followed by ZnT5 and ZnT1. For the ZIP family, ZIP7 was the most abundantly expressed, followed by ZIP14, ZIP6, ZIP1, ZIP10, and ZIP3A.

The expression levels of ZnT3, ZnT4, ZIP1, ZIP2, ZIP3A, ZIP5, ZIP7, and ZIP10 all fell 24 h after Cd exposure, but the expression of ZnT1 rose markedly.

In the same study, the expression profiles of ZnT and ZIP in UROtsa cells treated with buthionine sulfoximine, an inhibitor of glutathione (GSH) synthesis, were compared with those cells treated with 5-aza-2′-deoxycytidine, a DNA methylation inhibitor. The results indicated that Cd-induced ZnT1 expression was influenced mostly by GSH levels. Similarly, the effects of Cd on ZnT5, ZnT7, and ZIP8 expression levels appeared to vary with cellular redox status but not with DNA methylation status. However, the effects of Cd on ZIP14 gene expression seemed to depend on both GSH levels and the DNA methylation state.

Like UROtsa cells, the homeostasis of Zn in myeloid cells appeared to be regulated by DNA methylation [124]. Another study noted that the relationships of ZIP and ZnT transporters were highly sophisticated [125]. In summary, the experimental data described above suggested that the redox state and epigenic mechanisms regulate the expression of Zn transport proteins and cellular Zn homeostasis.

## 4. Global Health Threat of Environmental Cadmium

This section highlights results from cross-sectional and longitudinal cohort studies, showing the overall health impact of Cd at the exposure levels below the tolerable guidelines that presently exist. Special attention is given to the impact of Cd on eye disease, bones, and evidence for the beneficial health effects of Zn.

### 4.1. Cadmium and the World’s Leading Causes of Death

In the WHO global health report, cardiovascular disease (CVD), particularly ischemic heart disease, was the world’s top cause of death, which accounted for 16% of total global deaths in 2019 (https://www.who.int/news-room/fact-sheets/detail/the-top-10-causes-of-death) (accessed on 1 May 2024). The second, third, and ninth leading causes of death in 2019 were stroke, chronic obstructive pulmonary disease (COPD), and diabetes, respectively. Death from kidney disease rose from the 13th in 2000 to the 10th in 2019.

Verzelloni et al. (2024) conducted a dose–response meta-analysis of data from 26 studies, published from 2005 to 2023, to evaluate the contribution of Cd exposure to the risk of CVD [126]. They found that the risks of heart failure, coronary heart disease, and stroke were all increased with Cd exposure in a dose-dependent manner. Specifically, a blood Cd level of 1 μg/L and urinary Cd excretion rate of 0.5 μg/g creatinine appeared to be sufficient to increase the risk of having CVD by 2.58-fold and 2.79-fold, respectively.

Epidemiological studies implicating Cd exposure as a significant contributor to total global mortality and morbidity can be found in Table 4.

In the U.S. hypertension cohort [127], blood Cd levels ≥ 0.80 μg/L were associated with 1.85-, 1.76-, and 3.41-fold increases in the mortality from any causes, CVD, and Alzheimer’s disease, respectively. Cd appeared to increase CVD mortality markedly (OR 2.12) among non-smokers who had hypertension [127]. In the other two U.S. diabetes and CKD cohorts, urinary Cd levels > 0.60 μg/L were associated with a 49% increase in all-cause mortality among those with diabetes [128], while urinary Cd levels ≥ 0.60 μg/g creatinine were associated with a 75% increase in deaths from any cause among those with CKD [129].

In a cohort of Swedish women [130], Cd exposure was associated with 38% and 20% increases in risk of death from any cause and having bone fracture, respectively. These results were obtained when the top tertile of urinary Cd (median 0.54 µg/g creatinine) was compared with the bottom tertile (median urinary Cd of 0.20 µg/g creatinine).

In a Taiwanese cohort of the general population [131], there was a 35% increase in all-cause mortality per a 1 μg/L increment in urinary Cd. Also, in this study, death from any cause rose 35% for every 1 μg/dL increment of urinary Cu. The association between mortality and urinary Cu in the Taiwanese cohort study may at least reflect kidney Cd toxicity, given that Cu is a redox active metal and an ROS generator [131]. Furthermore, an increased urinary excretion of Cu among Cd-exposed subjects has been noted in studies from Japan [134,135], Thailand [136], and Korea [137]. Urinary Cu was associated with a reduced eGFR in Mexican children [138].

In a Taiwanese COVID-19 cohort [132], urinary Cd levels > 2.05 μg/g creatinine were associated with a 5.35-fold increase in the likelihood of suffering from severe symptoms from infections with COVID-19 [132].

In a Chinese COPD cohort [133], Cd exposure was found to contribute significantly to disease morbidity and mortality. The likelihood of having acute exacerbation and death rose, respectively, 2.26-fold and 2.12-fold per a 1 µg/L rise in blood Cd with adjustment for potential confounders and co-morbidity factors.

### 4.2. Public Health Significance of Environmental Cadmium

As data in Table 4 indicate, chronic exposure to Cd contributed substantially to mortality and the morbidity among those with hypertension, CVD, diabetes, CKD, COVID-19 infections, and COPD. These Cd exposure levels impacting mortality and morbidity, inferred from blood and urinary Cd levels, were low, and these exposure levels were found in a significant proportion of the populations, documented below.

A study from Thailand reported the percentage of Cd excretion ≥ 1 µg/g creatinine among non-smoking women who had low body iron stores to be 22.5% [139]. In the U.S. general population with an overall mean urinary Cd excretion of 0.5 µg/g creatinine, 2.5%, 7.1%, and 16% of non-smoking women (aged ≥20 years) had urinary Cd levels > 1, >0.7, and >0.5 μg/g creatinine, respectively [140]. Thus, the proportion of U.S. women at risk of adverse health effects of Cd was not negligible. The estimated mean Cd intake among U.S. women (n = 1002, mean age 63.4) was 10.4 μg/day, and the mean urinary excretion rate was 0.62 μg/g creatinine [141]. These dietary exposure and urinary Cd excretion levels were 21% and 12% of the tolerable intake level of 49.8 µg/day for an average 60 kg female (0.83 µg/kg body weight/day) and a urinary Cd threshold level of 5.24 µg/g creatinine, respectively.

The current population exposure to environmental Cd presents global public health significance and many challenges because metal Cd is detectable in virtually all food types, especially in those frequently consumed in large quantities such as rice, potatoes, wheat, leafy salad vegetables, and other cereal crops. The current environmental Cd exposure has now reached toxic levels in a significant proportion of many populations. Alarmingly, CKD is predicted to become the fifth leading cause of years of life lost by 2040 should an upward trend in its prevalence continue [142].

### 4.3. Evidence for Mitigative Effects of Zinc

Zn intake levels found to be associated with reduced health risks due to Cd exposure are summarized together with results from a high-dose Zn supplementation trial, known as Age-Related Eye Disease Study (AREDS). The AREDS supplement formulation included high-dose antioxidants [vitamin C (500 mg), vitamin E (400 IU), β-carotene (25 mg, equivalent to vitamin A 25,000 IU)] and high-dose zinc (80 mg) with additional 2 mg Cu to prevent Cu deficiency anemia from high-dose zinc [90].

#### 4.3.1. Cancer, COPD, and CKD

In a U.S. population study (NHANES 1988–1994), a Zn intake below the RDA was associated with an elevated body burden of Cd in both men and women, assessed with urinary Cd levels [143]. There was a 1.55-fold increase in death from cancer in women who consumed Zn below the RDA, compared with women who met the RDA [143]. In the same NHANES dataset [144], there was a 1.89-fold increase in the risk of COPD at a Zn intake below 8.35 mg/day, compared with a Zn intake >14.4 mg/day. The risk of COPD rose 3.48-fold at urinary Cd levels ≥ 0.79 μg/g creatinine [144]. A dietary Zn intake of 15 mg/day, which is higher than the RDA of 11 mg/day for men and 8 mg/day for women (Table 1), would be required to reduce the excessive COPD risk imposed by Cd exposure.

An increased risk of having a low eGFR was linked with serum Zn levels < 74 μg/dL and blood Cd levels > 0.53 μg/L [145]. A dietary Zn of 16.46 mg/day was sufficient to reduce the adverse effect of Cd on kidneys (low GFR) among participants in the NHANES 2003–2018 (n = 37,195), where an overall mean Zn intake level was 11.85 mg/day. In this study, Zn intake levels showed a U-shaped dose–response relationship with CKD risk [146], meaning that the Zn intake should not be too low (≤6.64 mg/day) or too high (>16 mg/day). Accordingly, dietary Zn levels of 15–16 mg/day, higher than RDA values, may help reduce excessive health risks due to Cd exposure.

#### 4.3.2. Cadmium, Macular Degeneration, and High-Dose Zinc Supplementation

Macular degeneration (MD) is the leading cause of blindness in adults, aged ≥ 50 years [147,148]. The dysfunction and death of the retinal pigment epithelium (RPE), which forms a blood–retinal barrier, are involved in the pathogenesis of MD [148,149,150]. Dysregulated metal homeostasis, evident from reduced levels of Zn and Cu in the RPE and choroid complex, may also be a contributing factor [151]. Like data for UROtsa cells (Table 3), various ZIP and ZnT are expressed in adult human RPE cells and the ARPE-19 cell line, which provide entry routes for both Zn and Cd [152,153,154,155].

A 1.56-fold increase in the risk of MD among NHANES 2005–2008 participants aged ≥ 60 years was associated with blood Cd levels ≥ 0.66 μg/L [156]. At urinary Cd as low as 0.35 μg/L, the risk of MD rose 3.31-fold in non-Hispanic whites, who appeared to be particularly susceptible to the ocular toxicity of Cd [157]. Similarly, studies from Korea observed 2.11-fold and 1.92-fold increases in the risk of MD among those with elevated Cd exposure, assessed with blood Cd levels [157,158].

Satarug et al. (2008) conducted a study using a human RPE cell culture model [159], and they reported that a 50–60% reduction in Cd accumulation was achieved after RPE cells were simultaneously exposed to Cd and Zn at a twofold higher molar concentration of Cd. A marked decrease in Cd accumulation was due possibly to Zn and Cd competition for the same influx transporters.

The result described above has provided a plausible explanation for reduced Zn levels in the RPE from MD patients [151] and the positive outcomes of high-dose Zn supplement in the U.S. AREDS [90,160]. In a 10-year follow-up analysis, no adverse effects were associated with the AREDS supplementation formulation, while the risks of developing advanced MD and moderate vision loss were decreased by 34% and 29%, respectively [160].

### 4.4. Cadmium, Zinc, and Bone Resorption

#### 4.4.1. Human Studies

The chronic ingestion of a high dose of Cd (>100 µg/day) is a known cause of itai-itai disease, marked by severe damage to kidneys and bones with multiple fractures due to osteoporosis and osteomalacia [161]. Notably, however, evidence for the adverse effects of Cd on the kidneys and bones have been observed in residents of a Cd-contaminated area of Mae Sot, Tak Province of Thailand, even though their Cd exposure levels were moderate [161,162,163,164]. Furthermore, as data in Table 4 indicate, the effects of Cd on the onset of osteoporosis and fracture have been observed in a prospective cohort of a population exposed to a very low level of Cd like Sweden [130].

An increased excretion of the biomarkers indicative of defective mitochondrial function has been found in studies of Mae Sot residents [165] and Swedish women [166], although exposure levels in Sweden were very low, compared to the residents of a Cd contamination region of Thailand. A bone effect in a Swedish study of Cd was observed in the absence of any effects on the kidneys [130]. This may reflect a very high sensitivity of bone to mitochondrial effects of Cd [167], leading to enhanced ROS production and oxidative damage. Bone tissues from patients with osteoporosis had higher Cd levels, compared to controls without osteoporosis [168].

#### 4.4.2. Experimental Studies

For osteoblast entry, Cd and Zn both utilize ZIP8 and ZIP14, which are expressed on the osteoblast surface [169]. Unlike Zn, however, no excretory mechanism exists for Cd, and Cd thus accumulates within osteoblasts [170]. Although this phenomenon has not been specifically investigated for bone cells, the induction of ZnT1 expression by Cd may result in the removal of Zn from bone cells, while Cd is retained (Section 3.1).

Ou et al. examined the effects of Cd on pre-osteoblasts (MC3T3-E1 subclone14 cell lines) at different concentrations for 60 h. They demonstrated that the presence of Cd was associated with lower alkaline phosphatase (ALP) secretion, attenuated cell viability, and a decreased mRNA expression of Runx2 and type I collagen [171]. Moreover, exposure to Cd led to DNA damage and induced apoptosis via the caspase-dependent pathway [171]. Zheng et al. cultivated osteoblasts obtained from fetal rat cranial bones with Cd at 1, 2, and 5 μM for two periods of time: 6 and 12 h. They showed that Cd decreased cell viability in a dose-dependent manner. The authors found that the rate of osteoblast apoptosis was 10% after exposure to 1 μM CdCl_2_, while this rate rose to 30% with 5 μM CdCl_2_ [172]. Exposure to Cd led to the release of mitochondrial cytochrome c, resulting in Cd-mediated apoptosis linked with the mitochondrial p53 signaling pathway. On the other side, Zn reversed most of the Cd effects and restored the viability of osteoblasts, the synthesis of ALP, collagen, and bone mass [173].

Tian et al. exposed the Saos-2 cell line to Cd as CdCl_2_ at 0.5, 1, 10, 20, 40, and 80 μM for 24–48 h. Their results indicated that Cd higher than 10 μM decreased cell viability. They also observed a notable change in the morphology of the osteoblast nuclei, characterized by bending, fragmentation, and eventual nuclear collapse at 20 μM [174].

Chen et al. conducted a study in which they exposed osteoblasts obtained from rat calvarias to various concentrations of Cd (0, 0.125, 0.5, and 2.0 mol/L) for 24 h, while osteoclasts obtained from the long bone of rats were exposed to different Cd concentrations (0, 0.03 mol/L) for 72 h. They found that Cd decreased the viability of osteoblasts and the activity of ALP in a dose-dependent manner. Moreover, the presence of Cd altered mineralization [175]. In another study, Cd at 0.03 mol/L potentiated RANKL expression which consequently increased the number of osteoclasts. In comparison, Zn downregulated RANKL/RANK expression, and osteoclastogenesis was diminished as a consequential result [176].

The mechanisms by which Cd induced osteoblast death were examined by Liu and colleagues. They utilized osteoblasts obtained from Sprague-Dawley rat fetuses treated with 0, 1, 2, and 5 μM Cd. They observed changes in osteoblast nucleus morphology and the upregulation of Bax but the downregulation of Bcl-2 (which has antiapoptotic activity) [177]. In another in vitro study utilizing MC-3T3-E1 cells, exposure to CdCl2 at 0–20 μM decreased cell viability, and promoted osteoblast apoptosis, which resulted from a decreased Bcl-2 level and accumulation of Bax mRNA and protein, and interfered with osteoblast formation by decreasing the expression of RANKL [178].

A diagrammatic representation of Cd-induced bone resorption is provided in Figure 3.

As depicted in Figure 3, Cd causes osteoblast premature cell death, and it stimulates the genesis of osteoclasts, thereby promoting bone resorption. Cd increases ZnT1 expression, leading the extrusion of Zn and Zn deficiency conditions while retaining Cd [168]. In comparison, Zn reduces osteoclast resorption activities and increases the number of osteoblasts [179,180].

In summary, it is evident from a series of experimental studies that Cd adversely affected osteoblast function. Cd and Zn appeared to impact the genesis of osteoclasts via the RANKL-RANK axis. In bones from rats fed with Zn deficient diet, a reduced osteoblastogenesis and an enhanced osteoclastogenesis were found to be a consequence of RANKL upregulation [181]. Because Cd and Zn interact with the same transport proteins, an increased dietary Zn intake could potentially mitigate some detrimental effects of Cd on bone primarily through decreasing the number of osteoclasts.

## 5. Conclusions

Most dietary Zn is absorbed through highly specific transport proteins, namely ZIP4, ZnT5, and ZnT1, but Zn–amino acid co-transport mechanisms may also be involved. In comparison, Cd is absorbed through many transport proteins such as ZIP14, DMT1, ATP7A, TRPV5, and TRPV6 which are necessary for the assimilation of essential metals Fe, Mn, Cu, and Ca. Furthermore, Cd complexed with MT (CdMT) and phytochelatinn(CdPC) can be absorbed through transcytosis and receptor mediated endocytosis. Consequently, the intestinal absorption rate of Cd can be as high as 45%.

As a mechanism to prevent toxicity from Zn overload, Zn is extruded from cells by ZnT1, an efflux transporter for Zn only. No equivalent exit route for Cd exists, and most acquired Cd is thus retained within cells.

Zn is involved in the fundamental biological mechanisms that maintain normal intermediary metabolism and cell function. It is vital for cellular redox signaling and antioxidant defense mechanisms. Its antioxidative function is through the activity of SOD, Nox, MT, HO-1, and bilirubin, all of which prevent oxidative damage from excessive ROS. Through its interaction with MTF-1, like Zn, Cd induces simultaneously MT and ZnT1, which can result in a reduction in Zn bioavailability, the disruption of cellular Zn homeostasis, and induced Zn deficiency conditions.

The current environmental Cd concentrations mean that many populations are now exposed to toxic levels of this metal, and there is compelling evidence that this is impacting global mortality and morbidity. The risks of heart failure, coronary heart disease, and stroke are all increased with Cd exposure in a dose-dependent manner. A blood Cd level of 1 μg/L and urinary Cd excretion rate of 0.5 μg/g creatinine are associated with an increased risk of CVD of 2.58-fold and 2.79-fold, respectively. Death from any causes among those with CKD or diabetes also rose with Cd exposure in a dose-dependent manner.

The main source of Cd exposure in non-smoking and non-occupationally exposed people is the food that is eaten. However, as the current dietary exposure guidelines are not low enough to reduce the risk of adverse health effects, public health measures should be instigated to help minimize Cd contamination in food chains. It would also be prudent to establish new health protective exposure guidelines. Ensuring adequate nutritional levels of essential metals, especially Zn, in populations at risk of Cd exposure is warranted.

## Figures and Tables

**Figure 1 biomolecules-14-00650-f001:**
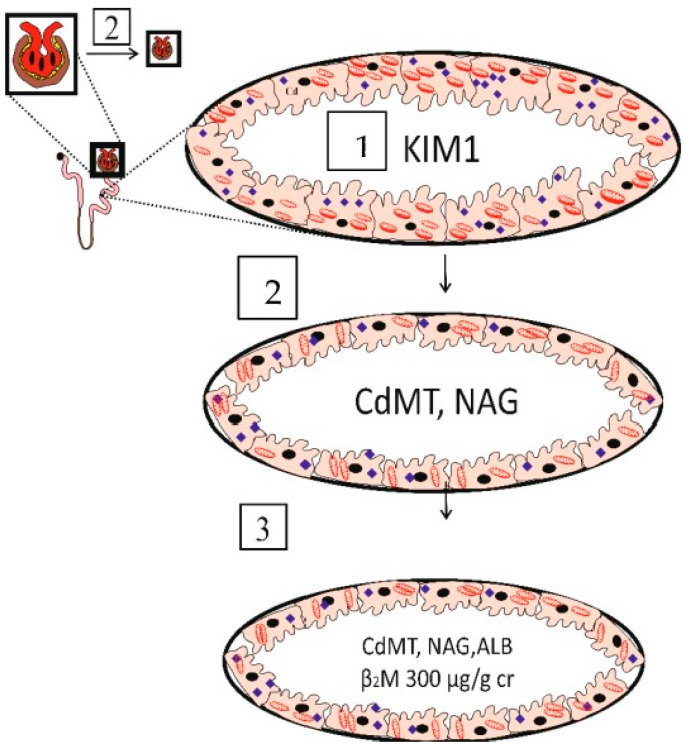
The tubulo-glomerular effects of cadmium. Phase 1, a subtle effect of Cd is indicated by the urinary excretion of KIM1. Phase 2, an intensified effect of Cd is indicated by a fall in the eGFR and urinary excretion of CdMT and NAG. Phase 3, overt effects of Cd on glomerular and tubular functions are indicated by a fall in the eGFR to 60 mL/min/1.73 m^2^ and a rise in urinary β_2_M excretion to 300 µg/g creatinine. Abbreviation: ALB, albumin, β_2_M, β_2_-microglobulin; CdMT, cadmium–metallothionine complexes; KIM1, kidney injury molecule 1; NAG, N-acetyl-β-D-glucosaminidase.

**Figure 2 biomolecules-14-00650-f002:**
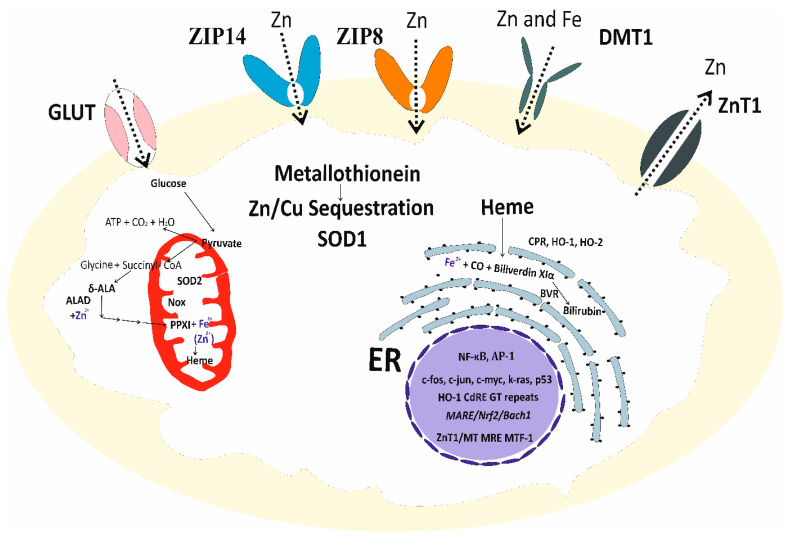
The cellular homeostasis of zinc and its antioxidative function. ZIP14, ZIP8, and DMT1 mediate Zn entry into cells. ZnT1 mediates Zn extrusion. The antioxidative function of Zn is through MT, SOD1, heme biosynthesis, and degradation, which supplies a substrate precursor for producing bilirubin. Abbreviations: GLUT, glucose transporter; SOD, superoxide dismutase; δ-ALA, delta-aminolevulinic acid; ALAD, aminolevulinic acid dehydratase; PPXI, protoporphyrin XI; ER, endoplasmic reticulum; CPR, cytochrome P450 reductase; HO-1, heme oxygenase-1; HO-2, heme oxygenase-2, BVR, biliverdin reductase; *CdRE*, *cadmium response element*; *MRE*, *metal response element*; MTF-1, metal response element-binding transcription factor-1.

**Figure 3 biomolecules-14-00650-f003:**
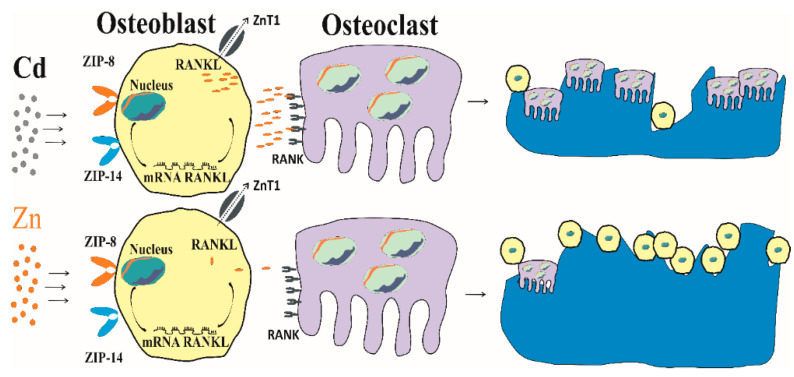
Bone resorption induced by cadmium (**upper** row) and maintenance of bone mass by zinc and (**lower** row). Effects of Cd and Zn are through RANKL-RANK axis. Abbreviations: Cd, cadmium; Zn, zinc; RANKL, receptor activator of nuclear factor kappa-Β ligand; RANK, receptor activator of nuclear factor kappa-Β.

**Table 1 biomolecules-14-00650-t001:** Recommended zinc intake levels versus tolerable intake levels of cadmium.

Nutrient/ Contaminant	Recommended Zinc Intake Level versus Tolerable Cadmium Level	Reference
Zinc (atomic weight 65.4)	An adult female (60 kg): 8 mg/day, or 0.13 mg per kg of body weight per day. An adult male (70 kg): 11 mg/day, or 0.16 mg per kg of body weight per day. Infants: 2–3 mg/day. Children: 5–9 mg/day. Pregnant women: 11–12 mg/day. Lactating women: 12–13 mg/day.	The U.S. National Academy of Sciences Institute of Medicine (IOM) [8]
Cadmium (atomic weight 112.4)	A tolerable lifetime intake of 2 g or an intake of 0.83 μg per kg body weight per day (58 µg per day for a 70 kg person). ^a^ Assumption: Absorption rate at 3–7%; urinary excretion of β_2_-microglobulin (β_2_M) levels ≥ 300 µg/g creatinine indicated kidney toxicity with a threshold level of 5.24 μg Cd/g creatinine.	JECFA [38]
Cadmium	BMDL or the NOAEL equivalent value of Cd exposure was 0.0104 µg/L of filtrate (≈0.01−0.02 µg/g creatinine). Assumption: Cd accumulation level producing a 5% decline in the eGFR represented the upper limit of a permissible Cd exposure level.	Satarug et al., 2022 [45]
Cadmium	A tolerable intake of 0.36 μg/kg body weight per day (25.2 µg per day for a 70 kg person) Assumption: Urinary β_2_M excretion levels ≥ 300 µg/g creatinine indicated kidney toxicity with a threshold level of 1 μg Cd/g creatinine	EFSA [46]
Cadmium	A tolerable intake level of 0.28 μg per kg body weight per day (16.8 µg per day for a 60 kg person). Assumption: Urinary β_2_M excretion levels ≥ 300 µg/g creatinine to indicate kidney toxicity with a threshold level of 3.07 μg Cd/g creatinine.	Qing et al., 2021 [47]
Cadmium	A tolerable intake level of 0.35 μg Cd per kg body weight per day, assuming a bone toxicity threshold level of 0.5 μg/g creatinine.	Leconte et al., 2021 [48]
Cadmium	By the reverse dosimetry PBPK model, tolerable intake levels range between 0.21 and 0.36 μg per kg body weight per day, assuming a similar bone and kidney toxicity threshold level of 0.5 μg/g creatinine.	Schaefer et al., 2023 [49]

PBPK, physiologically based pharmacokinetics; BMDL, benchmark dose limit; NOAEL, no-observed-adverse-effect level. ^a^ On a molar basis, a tolerable lifetime intake of Cd at 2 g is half of the total Zn in the body of 2 g. A lifetime intake of 1 g of Cd was linked to a 49% increase in the likelihood of dying from kidney failure [50].

**Table 2 biomolecules-14-00650-t002:** Specialized transport proteins for absorption of metal nutrients and cadmium.

Metal	Transporter	Localization	Description
Zn	SLC39A4 (ZIP4)	Apical membrane	Two zinc-binding sites in histidine-rich motif [64]. Transport dietary Zn into enterocytes. Functional loss due to mutation causes Zn deficiency [65].
Zn	SLC30A5 (ZnT5) variant B	Apical membrane	Transport Zn into enterocytes [66]. Zn supplementation depresses ZIP4 and ZnT5B protein levels in the ileum [67].
Zn	SLC30A1 (ZnT1)	Basolateral membrane	Transport Zn from cytoplasm to extracellular medium or intracellular vesicles [68,69]. Histidine-rich motif confers Zn selectivity over Cd. This Cd discrimination is unique to mammalian ZnT1 [69].
Fe, Mn, Cd	SLC39A14 (ZIP14)	Basolateral membrane	Trafficking Zn to tight junctions, especially in jejunum for maintenance of intestinal barrier [70,71].
Fe, Cd Cu, Zn	SLC11A2 (DMT1)	Apical membrane	Same high affinity for Cd as it has for Fe (Km 0.5~1 μM); high abundance in duodenum [72,73,74,75,76].
Fe, Zn, Co	SLC40A1 (FPN1)	Basolateral membrane	Transport Fe, Zn, and Co but not Cu, Cd, or Mn from basolateral membrane into portal blood [77,78].
Cu, Fe, Zn	SLC31A1 (CTR1)	Apical membrane, Early endosome	Transport dietary Cu into enterocytes [68,79].
Cu, Cd	ATP7A	Trans-Golgi network, Cytosol Basolateral membrane	Transport Cu into portal blood, and ATP7A mutations are associated with Menkes disease [68]. May contribute to Cd absorption [80].
Ca, Cd	TRPV5, TRPV6	Apical membrane	Transport Ca^2+^ into enterocytes [81] and may provide Cd entry routes into enterocytes [82,83].
Ca	Calbindin-D9k	Cytoplasm	Transport Ca to basolateral membrane and extrusion of Ca into portal blood [83,84,85]. Expression in ileum is induced by 1,25-dihydroxycholecalciferol [85].

SLC, solute-linked carrier; ZIP4, Zrt- and Irt-related protein 4; ZIP14, Zrt- and Irt-related protein 14; ZnT1; zinc transport 1; DMT1, divalent metal transporter 1; FPN1, ferroportin 1; CTR1, copper transporter 1; ATP7A, copper-transporting ATPases (Cu-ATPases); TRPV5, transient receptor potential vanilloid5; TRPV6, transient receptor potential vanilloid6.

**Table 4 biomolecules-14-00650-t004:** Effects of lifetime exposure to environmental cadmium on mortality and disease risks.

Study Population	Effects Observed and Cadmium Exposure Levels	Reference
United States NHANES, 1999–2012, n = 12,208 with hypertension Mean follow-up 10.8 years	Among those with hypertension, respective HRs (95% CI) for all-cause, CVD, and Alzheimer’s disease mortality were 1.85 (1.59, 2.14), 1.76 (1.33, 2.34), and 3.41 (1.54, 7.51), comparing blood Cd levels ≥ 0.80 versus ≤0.25 μg/L. HR (95% CI) for CVD mortality among non-smokers who had hypertension was 2.12 (1.36, 3.30)	Chen et al., 2023 [127]
United States NHANES, 2001–2010, n = 2945 with diabetes Mean follow-up period, 9.1 years	Among those with diabetes, HR (95% CI) for all-cause mortality was 1.49 (1.06, 2.09) at urinary Cd levels > 0.60 μg/L. HR (95% CI) for all-cause mortality was 1.65 (1.24, 2.19) at serum CRP levels > 0.49 mg/dL. HR (95% CI) for cancer mortality was 3.25 (1.82, 5.80) at serum CRP levels > 0.49 mg/dL.	Liu et al., 2022 [128]
United States NHANES, 1999–2014, n = 1825 with CKD Mean follow-up period, 6.8 years	Among those with CKD, HR (95% CI) for all-cause mortality was 1.75 (1.28, 2.39) at urinary Cd levels ≥ 0.60 μg/g creatinine. HR (95%CI) for death from any causes was 1.59 (1.17, 2.15) at blood Cd levels ≥ 0.70 μg/L.	Zhang et al., 2023 [129]
Sweden n = 4024 women, aged 56–85 years Mean follow-up 10.5 years	Respective HRs (95% CI) for all-cause mortality and any fracture were 1.38 (1.10, 1.74) and 1.20 (1.01,1.43), comparing top tertile of urinary Cd (median 0.54 µg/g creatinine) with bottom tertial (median urinary Cd of 0.20 µg/g creatinine).	Tägt et al., 2022 [130]
Taiwan n = 2497 (1001 males, 1496 females), Median follow-up 3.5 years.	HR (95%CI) for all-cause mortality was 1.35 (1.09, 1.68) per 1 μg/L increment of urinary Cd. HR (95%CI) for all-cause mortality was 1.35 (1.15, 1.58) per 1 μg/dL increment of urinary copper.	Liu et al., 2021 [131]
Taiwan n = 252 with severe COVID-19, n = 322 with non-severe COVID-19 June 2022–July 2023	Among subjects with COVID-19, OR (95%CI) for severe symptoms was 5.35 (1.12, 25.6) at urinary Cd levels > 2.05 μg/g creatinine.	Chiu et al., 2024 [132]
China n = 196 COPD patients Enrolment: September 2020–June 2022, Follow-up 2 years	Per every 1 µg/L increase in blood Cd, respective ORs (95% CI) for acute exacerbation, hospitalization, and mortality were 2.24 (1.17, 4.29), 1.28 (1.13, 1.56), and 1.69 (1.06, 2.70). After adjustment for potential confounders, respective ORs (95%CI) for acute exacerbation and mortality were 2.26 (1.03, 4.24) and 2.12 (1.13, 7.11) per 1 µg/L increment of blood Cd.	Sun et al., 2024 [133]

HR, hazard ratio; OR, odds ratio; CI, confidence interval; CRP, c-reactive protein; NHANES, National Health and Nutrition Examination Survey.

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
