# Peer review of "Modulation of Adverse Health Effects of Environmental Cadmium Exposure by Zinc and Its Transporters"

_biomolecules, 2024, doi:10.3390/biom14060650_

Round 1
Reviewer 1 Report
Comments and Suggestions for Authors
The review " Modulation of the Adverse Health Effects of Environmental Cadmium Exposure by Zinc and Its Transporters " by Cirovic et al. focuses on the impact of cadmium on human health.
The review highlights the increased mortality and various diseases associated with cadmium exposure, even at very low doses. Additionally, the authors discuss the correlation between zinc deficiency and cadmium toxicity. By interfering with zinc transporters, cadmium perturbs zinc homeostasis. In this review, the authors strongly suggest better control of cadmium levels in food, advocating for a lower “tolerable” intake level of cadmium. In addition, they propose that zinc supplementation might help prevent cadmium’s adverse effects, but caution should be taken as authors also mentioned that zinc excess can also be toxic and should be highly controlled.
Overall, the review is well-written and offers a comprehensive overview of current knowledge on cadmium toxicity and its link with zinc homeostasis. This review consolidates existing knowledge on the topic and could, in the future, help improve strategies to prevent cadmium’s adverse effects.
Here are a few points that could be modified before publication:
1- Can the authors specify in the text the reasons for cadmium contamination in food? Is it mainly due through anthropogenic activities? If so, could the authors refer to other reviews discussing this aspect to find ways to minimize cadmium contaminations of food chains?
2- Line 57-58: Can the authors give more details on the link between low zinc bioavailability and high amount of phytate?
3- Legend Figure 1: does the phase number correspond to the numbers depicted in the figure? It does not look like it, and it can be confusing.
4- Line 144: do the authors know if it is possible to measure KIM1 in the urine? Given that Phase 1 is reversible, it will be interesting to detect cadmium exposure at this stage before cadmium irreversibly affects the cells.
5- Line 285: revised the sentence. The word “in” should be removed.
6- Line 323-324 : This sentence should be reconsidered. The authors mentioned earlier that other metals, like copper, could pose problem with dietary zinc increase. The idea is really interesting but should be presented with caution.
7- Line 437: Reference “146 7”: is a coma is missing?
Author Response
REVIEWER 1
Comments and Suggestions
The review " Modulation of the Adverse Health Effects of Environmental Cadmium Exposure by Zinc and Its Transporters " by Cirovic et al. focuses on the impact of cadmium on human health.
The review highlights the increased mortality and various diseases associated with cadmium exposure, even at very low doses. Additionally, the authors discuss the correlation between zinc deficiency and cadmium toxicity. By interfering with zinc transporters, cadmium perturbs zinc homeostasis. In this review, the authors strongly suggest better control of cadmium levels in food, advocating for a lower “tolerable” intake level of cadmium. In addition, they propose that zinc supplementation might help prevent cadmium’s adverse effects, but caution should be taken as authors also mentioned that zinc excess can also be toxic and should be highly controlled.
Overall, the review is well-written and offers a comprehensive overview of current knowledge on cadmium toxicity and its link with zinc homeostasis. This review consolidates existing knowledge on the topic and could, in the future, help improve strategies to prevent cadmium’s adverse effects.
RESPONSE: Thank you for a thorough review of our work, insightful comments, and helpful suggestions.
Here are a few points that could be modified before publication:
Comment 1. Can the authors specify in the text the reasons for cadmium contamination in food? Is it mainly due through anthropogenic activities? If so, could the authors refer to other reviews discussing this aspect to find ways to minimize cadmium contaminations of food chains?
RESPONSE: Thank you for raising this important issue. We have inserted below paragraph (lines 89-98) to address the issue raised.
The three main reasons for persistent presence of cadmium in the environment are that over the years it has been widely used in many industrial processes, it is a byproduct of many mining activities, and it is frequent contaminant of cheap low-grade phosphate fertilizers still used in developing countries [39-42].
The levels of Cd in foods are still likely to gradually increase over time because as a metal it does not degrade, and it is readily taken up from the soil by food crops. Thus, whilst efforts should focus on reducing environmental contamination, ways to reduce the accumulation of Cd by plants and methods to enhance dietary Zn bioavailability are worth pursuing [43,44].
[39] Verbeeck, M.; Salaets, P.; Smolders, E. Trace element concentrations in mineral phosphate fertilizers used in Europe: A balanced survey. Sci. Total Environ. 2020, 712, 136419.
[40] Zarcinas, B.A.; Pongsakul, P.; McLaughlin, M.J.; Cozens, G. Heavy metals in soils and crops in Southeast Asia. 2. Thailand. Environ. Geochem. Health 2004, 26, 359-371.
[41] Zou, M.; Zhou, S.; Zhou, Y.; Jia, Z.; Guo, T.; Wang, J. Cadmium pollution of soil-rice ecosystems in rice cultivation dominated regions in China: A review. Environ. Pollut. 2021, 280, 116965.
[42] McDowell, R.W.; Gray, C.W. Do soil cadmium concentrations decline after phosphate fertiliser application is stopped: A comparison of long-term pasture trials in New Zealand? Sci. Total Environ. 2022, 804, 150047.
[43] Zhao, S.; Zhang, Q.; Xiao, W.; Chen, D.; Hu, J.; Gao, N.; Huang, M.; Ye, X. Comparison of transcriptome differences between two rice cultivars differing in cadmium translocation from spike-neck to grain. Int. J. Mol. Sci. 2024, 25, 3592.
[44] Scavo, S.; Oliveri, V. Zinc ionophores: chemistry and biological applications. J. Inorg. Biochem. 2022, 228, 111691.
Comment 2. Line 57-58: Can the authors give more details on the link between low zinc bioavailability and high amount of phytate?
RESPONSE: Effects of dietary phytate on Zn absorption are provided (lines 205-211) as quoted below.
Phytate (myo-inositol hexabisphosphate), Ca and proteins are three main food components that affect dietary Zn bioavailability [5]. Phytate chelates Zn, and Ca fortification of phytate-rich soy milk has been shown to overcome the low Zn availability. By preventing Zn-Ca phosphate coprecipitation, caseinophosphopeptides increased Zn bioavailability in phytate-rich diets [87]. Protein intake may in-crease Zn absorption through Zn-amino acid co-transport mechanisms [5]. The intestinal absorption may be enhanced also by Zn ionophores [44, 88].
[5] Hall, A.G.; King, J.C. The molecular basis for zinc bioavailability. Int. J. Mol. Sci. 2023, 24, 6561.
[44] Scavo, S.; Oliveri, V. Zinc ionophores: chemistry and biological applications. J. Inorg. Biochem. 2022, 228, 111691.
[87] Feng, Y.; Zhu, S.; Yang, Y.; Li, S.; Zhao, Z.; Wu, H. Caseinophosphopeptides overcome calcium phytate inhibition on zinc bioavailability by retaining zinc from coprecipitation as zinc/calcium phytate nanocomplexes. J. Agric. Food Chem. 2024, 72, 4757-4764.
[88] Pérez de la Lastra, J.M.; Andrés-Juan, C.; Plou, F.J.; Pérez-Lebeña, E. Theoretical three-dimensional zinc complexes with glutathione, amino acids and flavonoids. Stresses 2021, 1, 123-141.
Comment 3. Legend Figure 1: does the phase number correspond to the numbers depicted in the figure? It does not look like it, and it can be confusing.
RESPONSE: Phase numbers 1 and 2 in Figure 1 have been changed to match with the description in its legend.
Comment 4. Line 144: do the authors know if it is possible to measure KIM1 in the urine? Given that Phase 1 is reversible, it will be interesting to detect cadmium exposure at this stage before cadmium irreversibly affects the cells.
RESPONSE: A reference to study on urinary KIM1 has now been provided (lines 144-146) as quoted below.
In Taiwanese patients with CKD, urinary Cd concentrations correlated with KIM1, but not other conventional renal biomarkers [54]. Thus, KIM1 excretion could serve as an early warning sign of Cd toxicity and kidney pathology.
Tsai, K.F.; Hsu, P.C.; Lee, C.T.; Kung, C.T; Chang, Y.C.; Fu, L.M.; Ou, Y.C.; Lan, K.C.; Yen, T.H.; Lee, W.C. Association between enzyme-linked immunosorbent assay-measured kidney injury markers and urinary cadmium levels in chronic kidney disease. J. Clin. Med. 2021, 11, 156.
Comment 5. Line 285: revised the sentence. The word “in” should be removed.
RESPONSE: A paragraph has been rewritten to better explain zinc deficiency anemia (lines 274-279) as quoted below.
The activity of δ-aminolevulinic acid dehydratase (ALAD), an enzyme in heme biosynthesis is Zn dependent. Zn deficiency anemia is caused by insufficient amount of heme for hemoglobin synthesis [113]. Similarly, anemia due to lead (Pb) poisoning is attributable to a decreased heme biosynthesis because of Pb displacement of Zn in ALAD [114]. The activity of ALAD could also be decreased by Cd through its induction of MT, which eventually affects the formation of both heme and bilirubin.
Comment 6. Line 323-324: This sentence should be reconsidered. The authors mentioned earlier that other metals, like copper, could pose problem with dietary zinc increase. The idea is really interesting but should be presented with caution.
RESPONSE: The referred sentence was misplaced, and it has now been deleted. To better present a beneficial health effect of Zn supplementation, Section 4.4.2 have been rewritten (lines ) as quoted below.
4.4. Evidence for Mitigative Effects of Zinc
Zn intake levels found to be associated with a reduced health risks due to Cd ex-posure are summarized together with results from a high-dose Zn supplementation trial, known as Age-Related Eye Disease Study (AREDS). The AREDS supplement formulation included high-dose antioxidants [vitamin C (500 mg), vitamin E (400 IU), β-carotene (25 mg, equivalent to vitamin A 25,000 IU)], high-dose zinc (80 mg) with additional 2 mg Cu to prevent Cu deficiency anemia from high-dose zinc [90].
4.4.2. Cadmium and Macular Degeneration
Macular degeneration (MD) is a leading cause of blindness in adults, aged ≥ 50 years [148,149]. Dysfunction and death of the retinal pigment epithelium (RPE), which form a blood-retinal barrier, are involved in the pathogenesis of MD [149-151]. Dysregulated metal homeostasis, evident from reduced levels of Zn and Cu in the RPE and choroid complex may also be a contributing factor [152]. Like data for UROtsa cells (Table 3), various ZIP and ZnT are expressed in adult human RPE cells and ARPE-19 cell line, which provide entry routes for both Zn and Cd [153-156].
A 1.56-fold increase in the risk of MD among NHANES 2005–2008 participants aged ≥ 60 years was associated with blood Cd levels ≥ 0.66 μg/L [157]. At urinary Cd, as low as 0.35 μg/L, the risk of MD rose 3.31-fold in non-Hispanic whites, who appeared to be particularly susceptible to the ocular toxicity of Cd [158]. Similarly, studies from Korea observed 2.11-fold and 1.92-fold increases in risk of MD among those with an elevated Cd exposure, assessed with blood Cd levels [158,159].
Satarug et al. (2008) conducted a study using human RPE cell culture model [160], and they reported that a 50-60% reduction in Cd accumulation was achieved after RPE cells were simultaneously exposed to Cd and Zn at a twofold higher molar concentration of Cd. A marked decrease in Cd accumulation was due possibly through Zn and Cd competition for the same influx transporters.
The result described above has provided a plausible explanation for reduced Zn levels in RPE from MD patients [152] and the positive outcomes of high-dose Zn supplement in the U.S. AREDS [90, 161]. In a 10-year follow-up analysis, no adverse effects were associated with the AREDS supplementation formulation, while the risks of developing advanced MD and moderate vision loss were decreased 34% and 29%, respectively [161].
Comment 7. Line 437: Reference “146 7”: is a coma is missing?
RESPONSE: The number 7 was in error and it has been removed.

Reviewer 2 Report
Comments and Suggestions for Authors
This review discusses Zn's roles in cellular processes, the disruption caused by Cd, and the health impacts of Cd exposure, emphasizing the protective effects of Zn and the challenge of setting safe Cd intake levels
The article is a comprehensive review of zinc's essential roles and the impact of cadmium exposure, however, it suffers from several critical issues that require major revision to enhance clarity and coherence.
Some points are repeated unnecessarily, such as the role of Zn in cellular functions and the effects of Cd exposure.
Generally, the text contains redundant information and lacks sufficient context for some points.
In particular, it suffers from issues with clarity and coherence, as it presents an overwhelming amount of detail without clear transitions.
Grammar and syntax need improvement, as some sentences are awkwardly constructed and affect readability or grammatically incorrect and confusing.
In general the logical flow between argument is lacking, making it difficult to grasp the overall argument.
In several section the text lacks a clear central argument or focus, making it hard to discern the primary point being made.
The text would benefit from a clearer focus and more straightforward presentation of key points.
In conclusion, the review offers valuable insights into the roles of zinc and the impact of cadmium exposure on health. However, to enhance its effectiveness and readability, significant revisions are necessary to improve clarity, coherence, and the overall logical flow of the narrative.
Comments on the Quality of English Language
The text contains awkwardly constructed sentences that affect readability and comprehension. Some sentences are grammatically incorrect or confusing, detracting from the clarity of the message.
Author Response
REVIEWER 2
Comments and Suggestions
This review discusses Zn's roles in cellular processes, the disruption caused by Cd, and the health impacts of Cd exposure, emphasizing the protective effects of Zn and the challenge of setting safe Cd intake levels
The article is a comprehensive review of zinc's essential roles and the impact of cadmium exposure, however, it suffers from several critical issues that require major revision to enhance clarity and coherence.
RESPONSE: Thank you for reviewing our work, your comments, and suggestions for improvement. Changes to the text are in blue. To better reflect its content, we have rewritten the aims (lines 75-83), as quoted below.
This review aims to provide insights into the roles of Zn and the impact of Cd exposure on health. Firstly, it summarizes the RDA for Zn together with current Cd exposure guidelines and its nephrotoxicity threshold level. It discusses the specific metal transporters which are responsible for Zn and Cd absorption, and cellular acquisition of these metals. Secondly, it discusses the correlation between Zn deficiency and Cd toxicity, and by interfering with Zn transporters, Cd perturbs Zn homeostasis. Thirdly, it summarizes the overall health threat linked with environmental non-workplace Cd exposure, evident from prospective cohort studies. The protective roles of Zn and its transport proteins against Cd-induced bone resorption are highlighted.
Point 1: Redundancy and transition
Some points are repeated unnecessarily, such as the role of Zn in cellular functions and the effects of Cd exposure.
Generally, the text contains redundant information and lacks sufficient context for some points.
In particular, it suffers from issues with clarity and coherence, as it presents an overwhelming amount of detail without clear transitions.
RESPONSE: We have edited the text and removed repetitive sentences. Additional text has been added where necessary.
Point 2: Gramma and syntax
Grammar and syntax need improvement, as some sentences are awkwardly constructed and affect readability or grammatically incorrect and confusing.
RESPONSE: Grammar and syntax corrections have been undertaken. Problematic sentences have been replaced with new sentences.
Point 3:
In general, the logical flow between argument is lacking, making it difficult to grasp the overall argument.
In several section the text lacks a clear central argument or focus, making it hard to discern the primary point being made.
The text would benefit from a clearer focus and more straightforward presentation of key points.
In conclusion, the review offers valuable insights into the roles of zinc and the impact of cadmium exposure on health. However, to enhance its effectiveness and readability, significant revisions are necessary to improve clarity, coherence, and the overall logical flow of the narrative.
RESPONSE: Our paper has been revised extensively, and we hope it has satisfactorily addressed all concerns.
Comments on the Quality of English Language
The text contains awkwardly constructed sentences that affect readability and comprehension. Some sentences are grammatically incorrect or confusing, detracting from the clarity of the message.
RESPONSE: The English has been revised, and errors have been corrected. Rewording of sentences and have been undertaken where required.

Round 2
Reviewer 2 Report
Comments and Suggestions for Authors
After reviewing the revised manuscript with the tracked changes, I can confirm that the authors have adequately addressed the concerns and comments raised by the reviewer. Therefore, the manuscript could be accepted for publication.
Comments on the Quality of English Language--